# Inorganic sulfur fixation via a new homocysteine synthase allows yeast cells to cooperatively compensate for methionine auxotrophy

**Jason S. L. Yu**[1], **Benjamin M. Heineike**[1], **Johannes Hartl**[2], **Simran K. Aulakh**[1], **Clara Correia-Melo**[1], **Andrea Lehmann**[2], **Oliver Lemke**[2], **Federica Agostini**[2], **Cory T. Lee**[2], **Vadim Demichev**[2], **Christoph B. Messner**[1], **Michael Mülleder**[3], **Markus Ralser**[1,2]*

1 Molecular Biology of Metabolism Laboratory, The Francis Crick Institute, London, United Kingdom,
2 Department of Biochemistry, Charité Universitätsmedizin, Berlin, Germany, 3 Core Facility—High Throughput Mass Spectrometry, Charité Universitätsmedizin, Berlin, Germany

* markus.ralser@charite.de

**Data Availability Statement:** All data are available within the main text or the supplementary materials. The proteomic dataset used for the re-

## Abstract

The assimilation, incorporation, and metabolism of sulfur is a fundamental process across all domains of life, yet how cells deal with varying sulfur availability is not well understood. We studied an unresolved conundrum of sulfur fixation in yeast, in which organosulfur auxotrophy caused by deletion of the homocysteine synthase Met17p is overcome when cells are inoculated at high cell density. In combining the use of self-establishing metabolically cooperating (SeMeCo) communities with proteomic, genetic, and biochemical approaches, we discovered an uncharacterized gene product YLL058Wp, herein named Hydrogen Sulfide Utilizing-1 (*HSU1*). Hsu1p acts as a homocysteine synthase and allows the cells to substitute for Met17p by reassimilating hydrosulfide ions leaked from *met17Δ* cells into O-acetyl-homoserine and forming homocysteine. Our results show that cells can cooperate to achieve sulfur fixation, indicating that the collective properties of microbial communities facilitate their basic metabolic capacity to overcome sulfur limitation.

### Author summary

Sulfur perturbation activates a dormant hydrogen sulfide fixation route via a novel homocysteine synthase Hsu1p.

## Introduction

Despite decades of effort, a large proportion of the coding sequences expressed from eukaryotic genomes lack complete functional annotation [1–4]. One contributing factor to this situation is that many laboratory experiments are conducted under a limited set of growth conditions that do not fully represent the range of conditions organisms are exposed to in

analysis of the met17Δ phenotype can be accessed PRIDE with the dataset identifier PXD031160.

**Funding:** This work was supported by the Wellcome Trust IA grant (IA 200829/Z/16/Z to MR), the Bundesministerium für Bildung und Forschung (BMBF) MSCoresys grant (031L0220 to MR) the European Commission CoBiotech project Sycolim (ID#33 to MR), European Commission Horizon 2020 research grant (ERC-SyG-202 951475 to MR), the Francis Crick Institute which receives its core funding from: Cancer Research UK (FC001134 to MR), UK Medical Research Council (FC001134 to MR) and the Wellcome Trust (FC001134 to MR) and the Swiss National Science Foundation Postdoc Mobility Fellowship 191052 to JH) The funders had no role in study design, data collection and interpretation, or the decision to submit the work for publication.

**Competing interests:** The authors have declared that no competing interests exist.

**Abbreviations:** EDTA, ethylenediaminetetraacetic acid; HSU1, Hydrogen Sulfide Utilizing-1; LC–MS, liquid chromatography mass spectrometry; LC-SRM, liquid chromatography-selected reaction monitoring; LIC, ligation independent cloning; MWCO, molecular weight cutoff; Ni-NTA, nickel-NTA; OAHS, O-acetylhomoserine; OAS, O-acetylserine; PLP, pyridoxal 5′-phosphate; SeMeCo, self-establishing metabolically cooperating community; SC, synthetic complete; SM, synthetic minimal; WT, wild-type; YNB, yeast nitrogen broth.

nature [5]. The budding yeast *Saccharomyces cerevisiae* is a key model organism for the discovery and characterization of gene function in eukaryotes. Moreover, because of its importance in biotechnology, the response to varying carbon and nitrogen sources has been extensively studied in this organism [6,7]. However, although the utilization of sulfur is an equally fundamental process, other than the core genes that are involved in the assimilation and utilization of inorganic sulfur into metabolically useful organosulfur compounds, its broader genetic network remains less well understood [8–10].

In many eukaryotes, sulfur is primarily assimilated in the form of sulfate ($SO_4^{2-}$), which becomes successively reduced to sulfide ($S^{2-}$). A key enzyme in the eukaryotic sulfur assimilation process is homocysteine synthase, which, in *S. cerevisiae*, is encoded by the *MET17* gene (also known as *MET15* or *MET25)*. Met17p catalyzes the fixation of inorganic sulfide with O-acetylhomoserine (OAHS) to form homocysteine (Fig 1A). Homocysteine is subsequently converted to methionine via the methionine synthase (Met6p) or into cysteine via cystathionine-β-synthase and cystathionine-γ-lyase (Cys4p and Cys3p, respectively) [11–13]. Homocysteine therefore provides the central organosulfur pool from which the sulfur-bearing amino acids methionine and cysteine are synthesized [14,15].

Deletion of *MET17* (*met17Δ*) renders cells auxotrophic for organosulfur compounds such as methionine, cysteine, homocysteine, or S-adenosyl methionine [15–18]. This led to the widespread use of the *met17Δ* allele as an auxotrophic marker for genetic experiments [19]. These alleles were crossed into many laboratory strains, including the W303 and S288C derivatives that gave rise to the yeast deletion collection [20]. However, *met17Δ* yeast exhibits an atypical growth phenotype, and cells overcome the auxotrophy under specific conditions. When *met17Δ* cells are replicated as thick patches, they continue growth even in the absence of an organosulfur source. The phenomenon occurs in a temperature-sensitive manner, with growth of the auxotrophs being most evident at 22°C and decreasing linearly with temperature up to 37°C [21]. This cell density and temperature sensitive phenotype has also been described in distantly related yeasts like *Candida albicans* and *Yarrowia lipolytica* [22,23] and is hence not a species-specific phenomenon. Among the possible explanations for this paradoxical phenotype, Cost and Boeke [21] suggested that *met17Δ* could leak and share organosulfur metabolites between cells once they reached a critical cell density.

Here, we studied the biochemical basis of the *met17Δ* phenotype and discovered a previously overlooked metabolic bypass that explains the growth of *met17Δ* in absence of organosulfur compounds. We found that an uncharacterized protein, which we herein have named Hydrogen Sulfide Utilizing-1 (Hsu1p, YLL058Wp), encodes a metabolic enzyme that directly assimilates sulfur from hydrogen sulfide, thereby enabling cell growth by resolving methionine auxotrophy once critical concentrations of sulfide are leaked from *met17Δ* cells. We provide evidence that Hsu1p functions as a homocysteine synthase, performing the fixation of inorganic sulfide in place of Met17p, thereby generating the homocysteine required for methionine and cysteine biosynthesis.

## Results

### The conditional nature of organosulfur auxotrophy in *met17Δ* yeast

The ability of *met17Δ* yeast strains to overcome organosulfur auxotrophy at high cell density was first described by Cost and Boeke [19,21]. This phenotype is nonheritable, which ruled out adaptive, secondary mutations concurrent with *met17Δ* as the cause. We studied the phenotype in two haploid yeast strains in the BY4741 background [24]: (i) a prototrophic strain in which auxotrophies were repaired by genomic integration of the missing genes (prototroph or wild-type (WT): *HIS3*, *LEU2*, *URA3*, *MET17*); and (ii) an analogous strain in which the

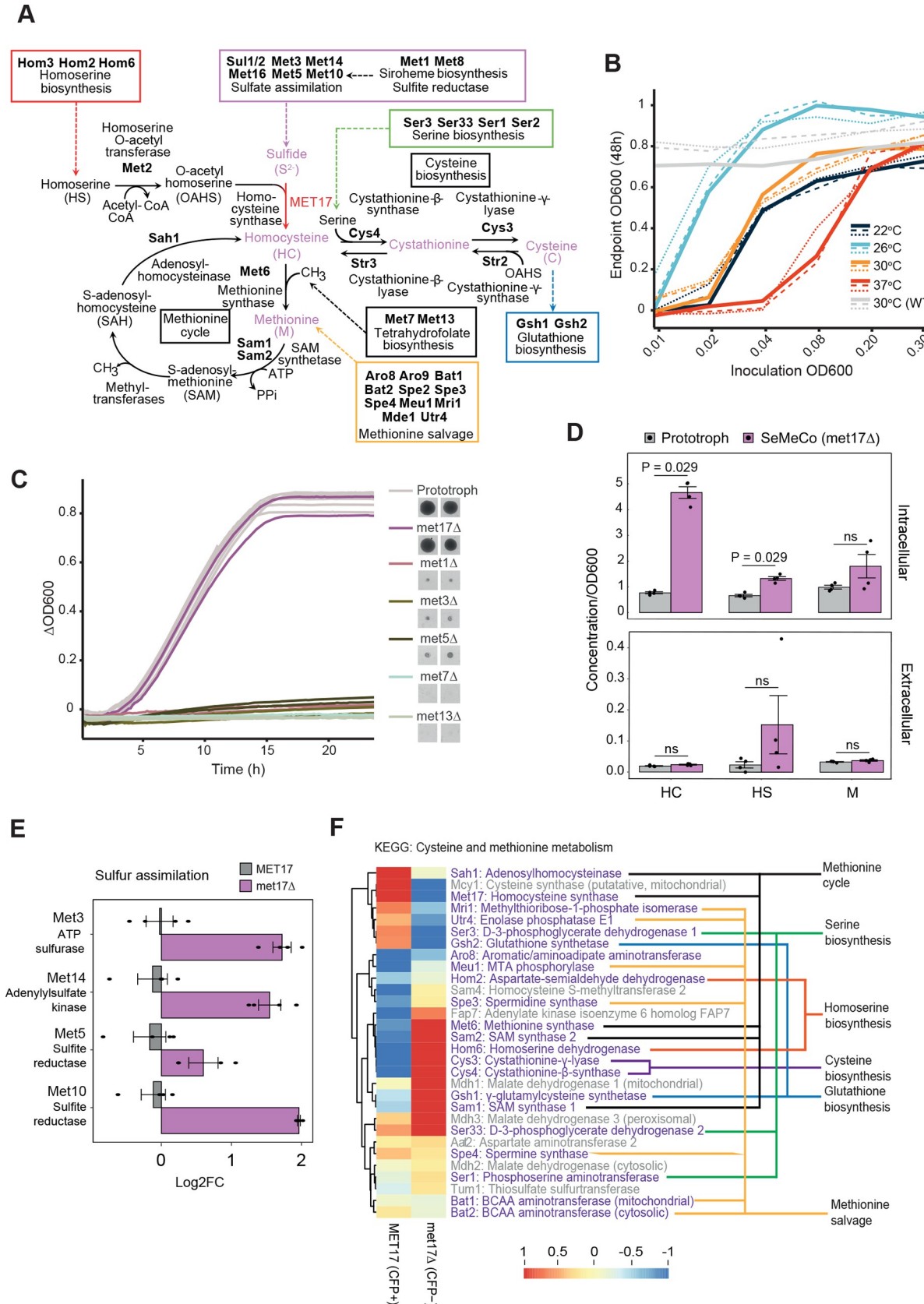

**Fig 1. Characterization of the *met17Δ* growth paradox by metabolomics and proteomics. (A)** The core and associated metabolic pathways that feed into and out of methionine and cysteine biosynthesis, including the gene names that encode for the respective enzymes in *S. cerevisiae*. Metabolites bearing sulfur are colored in light purple. **(B)** Liquid cultures of *met17Δ* yeast in minimal media without methionine inoculated at 6 different cell densities ($OD_{600nm}$ = 0.01 to 0.3) and cultured at 4 different temperatures (22, 26, 30, 37˚C) for 48 h. Gray lines correspond to the prototrophic control strain cultured at 30˚C. Lines represent three replicates per condition. **(C)** Growth curves of gene deletion strains implicated in methionine metabolism, as compared with a prototrophic strain in minimal medium after 24 h. Adjacent images depict growth of indicated deletion strains on solid minimal media after 72 h. **(D)** Metabolomic analysis and measurement of HC, HS, and M in extracellular and intracellular fractions between exponentially growing prototrophic and SeMeCo cultures normalized to the final $OD_{600nm}$. Data are from 4 biologically independent samples ($n$ = 4). Adjusted $p$-values from unpaired Wilcoxon rank sum test are as indicated; ns indicates no statistical significance. **(E)** Differential protein expression analysis of sulfur assimilation enzymes between sorted CFP+ (*MET17*) and CFP− (*met17Δ*), where $n$ = 4 biologically independent replicates. **(F)** KEGG pathway analysis and hierarchical clustering of sorted cells as in (E) of all detected enzymes involved in cysteine and methionine metabolism characterized by associated metabolic pathways (colored). The data underlying this figure can be found in S1 Data, with the exception of (E) and (F), which can be found in the PRIDE with the dataset identifier PXD031160. HC, homocysteine; HS, homoserine; M, methionine; PRIDE, Proteomics Identification Database; SeMeCo, self-establishing metabolically cooperating community.

*met17Δ* allele was not repaired (*met17Δ*: *HIS3*, *LEU2*, *URA3*, *met17Δ*). As previously observed by Cost and Boeke [19,21], our *met17Δ* strain was also able to grow in the absence of methionine over the course of 8 days when replicated in thick patches, independently confirming the reproducibility of this phenomenon (S1A Fig).

We started our investigations into the biochemical basis of this phenotype by ruling out amino acid contaminations in commercial yeast nitrogen broth (YNB), a common source of conflicting and unusual growth phenotypes. We validated the concentration of sulfur containing amino acids using liquid chromatography-selected reaction monitoring (LC-SRM) [25]. We detected levels of methionine, homocysteine, and glutathione at a concentration at least 100,000-fold lower than typical levels found in replete yeast media (S1B Fig). Therefore, the growth of *met17Δ* cells in the absence of methionine was not due to the presence of growth-relevant concentrations of sulfur-containing amino acids in the growth medium. To have a more experimentally tractable system to further our investigations, we next explored the conditions under which the growth phenotype was reproducible in liquid culture in a 96-well plate format. We inoculated the *met17Δ* strain into methionine-free medium at a range of initial densities ($OD_{600nm}$ = 0.01 to 0.3) and cultured under four temperatures (22, 26, 30, and 37˚C), recording the endpoint $OD_{600nm}$ at 24 h (S1C Fig) and 48 h (Fig 1B) as a measure of growth. At typical yeast culture temperatures (22, 26, and 30˚C), robust growth could be observed where the initial inoculation density was $OD_{600nm}$ > 0.02, in agreement with previous observations [19]. Growth at the lowest inoculation density ($OD_{600nm}$ = 0.01) was only observed when the culture temperature was at 26˚C, confirming that culture temperatures alter the cell density threshold that permits growth. Similarly, growth was permissive at 37˚C only when the initial cell density was high ($OD_{600nm} \geq 0.08$). In contrast, the WT strains could robustly grow at 30˚C irrespective of initial inoculation density (Fig 1B). Although we did not observe a strict correlation of temperature with growth as previously reported, growth was greatly enhanced at lower temperatures. Therefore, we established 25˚C as the standard culture temperature for all subsequent experiments, with low-density ($OD_{600nm}$ < 0.01) and high-density ($OD_{600nm} \geq$ 0.02) inoculums being nonpermissive and permissive densities for growth in the absence of organosulfur supplementation, respectively. These thresholds were independent of culture size (96-well versus 20 ml flasks; S1D Fig).

Next, to probe whether the cell density limitation is specific to *met17Δ* or whether it is a more general phenomenon related to methionine or amino acid auxotrophy, we compared the growth of the *met17Δ* strain to other enzyme knockout strains of sulfur assimilation pathways (*met1Δ*, *met3Δ*, *met5Δ*, *met7Δ*, or *met13Δ*) or amino acid metabolism (*argΔ*, *hisΔ*, *lysΔ*, *trpΔ*, and *aroΔ*). All auxotrophic strains demonstrated growth deficiency, but only the *met17Δ* formed biomass close to that of the WT strain (Figs 1C and S1E). This result suggested that the

ability to overcome the growth defect is not a common phenotype of methionine or amino acid auxotrophs but rather a specific phenotype that occurs upon the deletion of *MET17*. We speculated that either another enzyme replaces the function of Met17p in these conditions or else a metabolic shunt exists that allows yeast to circumvent the canonical sulfur utilization pathway to fuel growth.

## The ability to overcome organosulfur auxotrophy does not require the exchange of homocysteine or methionine

Cost and Boeke [21] speculated that *met17Δ* cells could overcome the growth defect through the sharing of organosulfur metabolites between cells to growth-relevant levels. Indeed, previous work from us and others has shown that communal yeast cells can effectively share metabolites to overcome auxotrophies [26,27]. In order to better understand the possible impact of metabolite exchange in the context of the ability of *met17Δ* cells to grow in the absence of methionine, we employed self-establishing metabolically cooperating (SeMeCo) communities, a synthetic yeast community designed to study metabolite exchange interactions between auxotrophs [27]. SeMeCo communities are composed of cells that express different combinations of auxotrophic markers and are established from a prototrophic founder cell through the stochastic loss of plasmids encoding one or more enzymes that compensate for metabolic deficiencies present in the genome. In the BY4741 background, these are *met17Δ*, *his3Δ*, *leu2Δ*, and *ura3Δ*, which induce methionine, histidine, leucine, and uracil auxotrophies, respectively. In SeMeCos, the degree of metabolite sharing influences the frequency of a respective auxotroph in the population. Examining data recently acquired by us revealed that *met17Δ* cells are the most dominant subpopulation within exponentially growing SeMeCo cultures (approximately 60%), and more frequent than cells harboring the three other metabolic deficiencies (S1F Fig) [28]. To understand the role that metabolite sharing plays in overcoming the loss of *MET17* in the SeMeCo system, we recorded the metabolic profiles of SeMeCos by LC-SRM and compared these against the metabolic profiles of prototrophic communities. In SeMeCo communities dominated by *met17Δ* cells, we detected a relative increase in intracellular homocysteine and homoserine concentrations compared to prototrophic communities (Fig 1D, intracellular). However, extracellular concentrations of homocysteine and methionine were not statistically different, although homoserine levels were elevated upon the exclusion of an outlying measurement (Fig 1D, extracellular). This result suggested that indeed, the *met17Δ* metabolic deficiency might be overcome through the exchange of an upstream precursor rather than directly via methionine sharing.

## *met17Δ* cells up-regulate enzymes involved in sulfur assimilation, cysteine and methionine metabolism

We have previously recorded proteomes from auxotrophs and prototrophs isolated from SeMeCo cultures using fluorescence activated cell sorting, and a CFP marker expressed from the same plasmid as the auxotrophic marker ([28]; PRIDE project: PXD031160). A differential expression analysis comparing specifically the *MET17* and *met17Δ* subpopulations revealed a clear up-regulation of sulfur assimilation enzymes (Met3p, Met14p, Met5p, and Met10p) and enzymes of the methionine salvage pathway (Meu1p, Spe3p, Spe4p, Bat1p, and Bat2p) in the *MET17* strain (Fig 1F). This result is consistent with a typical feedback response to the lack of methionine. Counterintuitively, however, several of the downstream enzymes, including Met6p and Gsh1p, that should carry no flux in the absence of Met17p, had increased protein levels (Fig 1A and 1F). In addition, the abundance of several core enzymes of the methionine cycle (Met6p, Sam1p, Sam2p), serine biosynthetic pathway (Ser1p, Ser33p), and cysteine

biosynthetic pathway (Cys3p, Cys4p) were also increased, although this was not observed with all enzymes (Sah1p, Ser3p). This data argued that homocysteine could be further metabolized despite the absence of Met17p, which would suggest the existence of a metabolic bypass to the homocysteine synthase activity of Met17p.

## Hydrogen sulfide fixation drives growth in the absence of methionine

We speculated that the intracellular accumulation of homocysteine in *met17Δ* cells ([Fig 1D]) could indicate the presence of an enzyme-catalyzed reaction that forms homocysteine independent of Met17p. As sulfide overflow is a defining characteristic of *met17Δ* cells [19,21], one possible explanation would be the presence of an enzyme that can utilize sulfide or its protonated forms (hydrosulfide ($HS^-$) and hydrogen sulfide ($H_2S$)) to incorporate sulfur and form homocysteine. To test this hypothesis, we exploited the cell density dependency of *met17Δ* cells in liquid culture that leads to the resolution of organosulfur auxotrophy as a readout (Figs [1B] and [S1C and S1D]). When conditioned medium from *met17Δ* cells cultured either at high-density ($OD_{600nm} > 0.02$, permissive) or low-density ($OD_{600nm} < 0.01$, nonpermissive) was filtered and inoculated with fresh cells at low density, only the filtered medium from the high-density culture could support cell growth. Moreover, this property was abolished upon precipitation of $H_2S$ with lead acetate ([Fig 2A]). To further substantiate whether $H_2S$ could be responsible for the growth, we supplemented two low-density cultures with sodium hydrosulfide (NaHS) that dissociates to form hydrosulfide ions and, subsequently, both aqueous and gaseous forms of $H_2S$ ([Fig 2B], reactions (ii-iv)). Taking advantage of the phase equilibria of $H_2S$, one culture was left unsealed to drive the equilibria towards the formation and dissipation of gaseous $H_2S$ from the system ([Fig 2B], reaction (iv)). Growth was only observed in the sealed culture, consistent with the assumption that a critical concentration of $H_2S$ is required to overcome the *met17Δ* auxotrophy ([Fig 2C]). In parallel, we manipulated the phase equilibria in the opposite direction, allowing the accumulation of aqueous $H_2S$ in low-density cultures via outgassing from either a high-density culture or a concentrated solution of NaHS by connecting the vessels with a small piece of rubber tubing to create a closed system. In both situations, growth was observed in the low-density culture ([Fig 2D]). To further substantiate that it is $H_2S$, which resolves the auxotrophy, we evaluated its presence and uptake across different cultures and growth conditions, exploiting the chemical properties of 7-azido-4-methylcoumarin (AzMC). In the presence of $H_2S$, AzMC undergoes selective reduction of the azido moiety to form 7-amino-4-methylcoumarin (AMC), which fluoresces when exposed to ultraviolet light [29]. When minimal media was compared with cultures inoculated at low density, no significant difference in AMC fluorescence was observed. Conversely, the fluorescence in NaHS-supplemented cultures decreased in the presence of cells, suggesting utilization of $H_2S$ and its concomitant depletion from the medium ([Fig 2E]). This fluorescence assay also indicated that high-density cultures had higher levels of aqueous $H_2S$, relative to low-density cultures, orthogonally confirmed by the precipitation of lead sulfide in high-density cultures ([Fig 2A]).

Finally, to test if sulfur does indeed transition from sulfate through to the biosynthesis of cysteine and methionine in the absence of *MET17*, we performed isotopic tracing with $S^{34}$-labeled ammonium sulfate (Figs [2F] and [S2A]). Uptake of this isotope label by sulfur assimilation via the up-regulation of the associated enzymes should lead to $S^{34}$ dissemination into all subsequent sulfur bearing derivatives via sulfide overflow (Figs [1A] and [S2B]). We quantified the ratios of $S^{32}$ and $S^{34}$ sulfur in methionine and reduced glutathione, the downstream products derived from homocysteine and cysteine biosynthesis, respectively, using liquid chromatography mass spectrometry (LC–MS). When native ($S^{32}$) ammonium sulfate was supplied, the assay detected $S^{34}$ at its expected environmental isotopic abundance ([S2B Fig]; 4% to 6%).

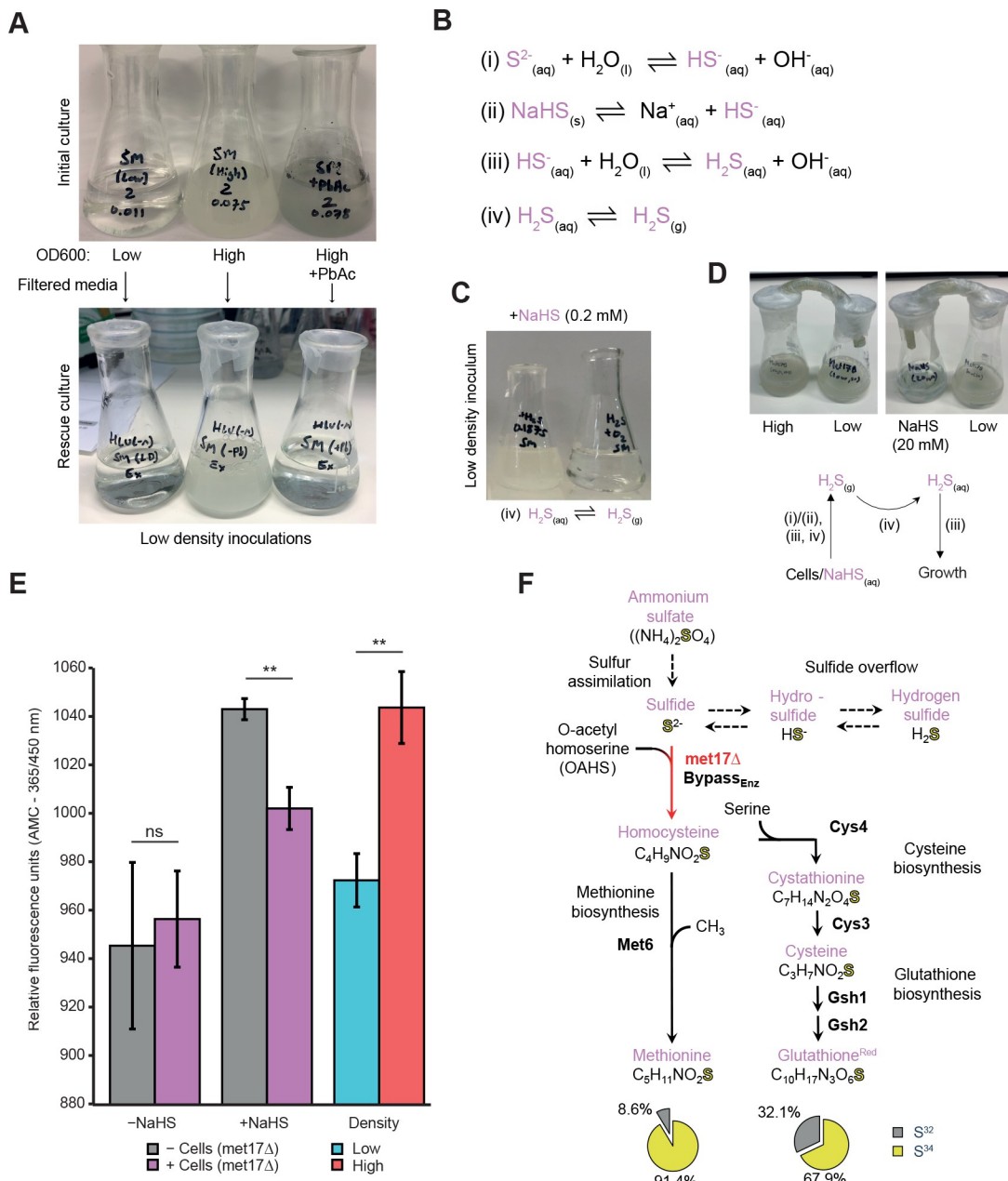

**Fig 2. Hydrogen sulfide is essential for the rescue of organosulfur auxotrophy of *met17Δ*. (A)** *met17Δ* cultures inoculated at low starting density ($OD_{600nm} < 0.01$) and high densities ($OD_{600nm} \geq 0.02$) (upper panel, left and middle). One flask of high-density inoculation was supplemented with Pb(II) acetate to precipitate $H_2S$, forming lead sulfide (upper panel, right). Conditioned media from these three cultures was subsequently filtered, replenished with YNB and glucose, and reinoculated at low density (lower panel). Data are representative from two biologically independent experiments. **(B)** Reaction schemes. Hydrolysis of sulfide is in equilibrium with hydrosulfide and hydrogen sulfide (i, iii, iv). NaHS can serve as a hydrosulfide donor (ii) in solution. Hydrogen sulfide preferentially undergoes phase transition to a gaseous state (iv) in unsealed vessels or establishes a dynamic equilibrium with its aqueous counterpart in sealed vessels. **(C)** Growth of *met17Δ* cultures supplemented with NaHS and inoculated at low density, sealed vs. unsealed. **(D)** Cultures inoculated with the *met17Δ* strain where the vessels are connected via rubber tubing to facilitate gaseous exchange, either between a high-density culture or a high-concentration solution of NaHS (20 mM). Schematic indicates the reactions occurring at each stage to facilitate the growth of low-density cultures. **(E)** Quantification of hydrogen sulfide production and uptake via conversion of AzMC to AMC [21] between unsupplemented and NaHS supplemented cultures inoculated at low density in the presence (purple) or absence (gray) of *met17Δ* cells. Blue and red bars indicate a separate experiment where hydrogen sulfide production was quantified *met17Δ* cultures inoculated at either low (blue) or high (red) density, in a similar experiment as shown in (A). *P* values calculated via two-tailed Student *t* test, where *n* = 3 independent replicates. **(F)** Isotopic tracing experiment with pathway map indicating the

biosynthetic transfer of isotopic $S^{34}$ from ammonium sulfate to methionine (left group) and glutathione (right group) via methionine and cysteine biosynthetic pathways, respectively. Pie charts indicate the percentage of $S^{34}$ for cultures supplemented with $S^{34}$-labeled ammonium sulfate. The data underlying this figure can be found in S1 Data. AMC, 7-amino-4-methylcoumarin; NaHS, sodium hydrosulfide; YNB, yeast nitrogen broth.

Conversely, when we supplied $S^{34}$-labeled ammonium sulfate, $S^{34}$-methionine (91.4%) and $S^{34}$-reduced glutathione (67.9%) accumulated to levels well above natural abundances, indicative of sulfur transfer from inorganic sulfate to organic sulfur-bearing metabolites. Thus, *met17Δ* assimilated inorganic sulfur from ammonium sulfate, and yeast cells are competent in the use of $H_2S$ as a source for methionine and cysteine biosynthesis.

## A targeted genetic screen identifies the bypass enzyme responsible for H$_2$S-mediated growth in the absence of methionine

Having established homocysteine formation via $H_2S$ fixation as the process by which methionine and cysteine auxotrophy could be overcome, we next used a targeted genetic approach to identify the enzyme responsible. We started by identifying strains in the yeast deletion collection that are associated with sulfur metabolism [20]. We selected 15 strains (S1 Table) that carried a deletion in the sulfur metabolism-associated gene in addition to deletions of four auxotrophic markers in their genome that include *MET17* (*his3Δleu2Δura3Δmet17Δ*) based on two criteria: (i) the deleted gene product was a part of either the sulfur assimilation or organosulfur biosynthetic pathways (S2B Fig); or (ii) the deleted gene product was identified as an ortholog to Met17p according to the eggNOG database [30–32] (Fig 1A and S2 Table). These selected deletion strains were cultured in synthetic drop-in media and tested for *met17Δ* growth in the presence of NaHS. In a background of *MET17* deletion, *sul1/2Δ*, *met3Δ*, *met14Δ*, *met5Δ*, and *met10Δ* strains demonstrated robust growth only in the presence of NaHS, while the *met6Δ* strain lacking the methionine synthase was unable to grow. These growth phenotypes indicated that $H_2S$ utilization and the resolution of organosulfur auxotrophy in the presence of NaHS was independent of sulfur assimilation, but dependent on sulfur utilization (Figs 3A and S2B and S2C). Critically, these experiments also revealed the dependency of the $H_2S$ utilization pathway on homoserine-O-acetyltransferase that is responsible for the conversion of homoserine to OAHS as the *met2Δ* strain was also unable to grow (Fig 3B). Since OAHS is a substrate shared by both Met17p and Str2p, this implies a similar catalytic mechanism (Fig 1A) [17]. Of the strains deficient in Met17p orthologs, the *met17Δcys3Δ* strain also could be rescued by NaHS supplementation, albeit with an extended lag phase, which suggests the existence of a direct route towards cysteine formation via $H_2S$, possibly via cysteine synthase activity (S2C and S3B Figs, gray pathway; [14,33,34]). Notably, the *str2Δ* and *str3Δ* strains also had a lengthened lag phase upon NaHS supplementation. This suggested that the degree of transsulfuration from cysteine to homocysteine may act to regulate the reaction efficiency, although neither Str2p nor Str3p appears to directly catalyze the bypass reaction itself. Intriguingly, we observed that *YLL058WΔ* led to the loss of the ability to utilize $H_2S$ for growth (Fig 3A). This defect was not observed in strains deficient in the other Met17p orthologs (*irc7Δ*, *YML082WΔ*, and *YHR112CΔ*), indicating that YLL058Wp could potentially function as the bypass enzyme. We next asked if the gene products of *STR2* and *YLL058W* could operate as a homocysteine synthase in place of Met17p to rescue organosulfur auxotrophy. Str2p has been shown to be a cystathionine-γ-synthase, catalyzing the conversion of cysteine to cystathionine using OAHS as a substrate (Fig 1A). When operating in tandem with Str3p, this permits the transsulfuration of cysteine to methionine [14,35]. Since Str2p shares the same substrate as Met17p, there was a possibility it could incorporate sulfur via a secondary or promiscuous reaction. The catalytic properties of our

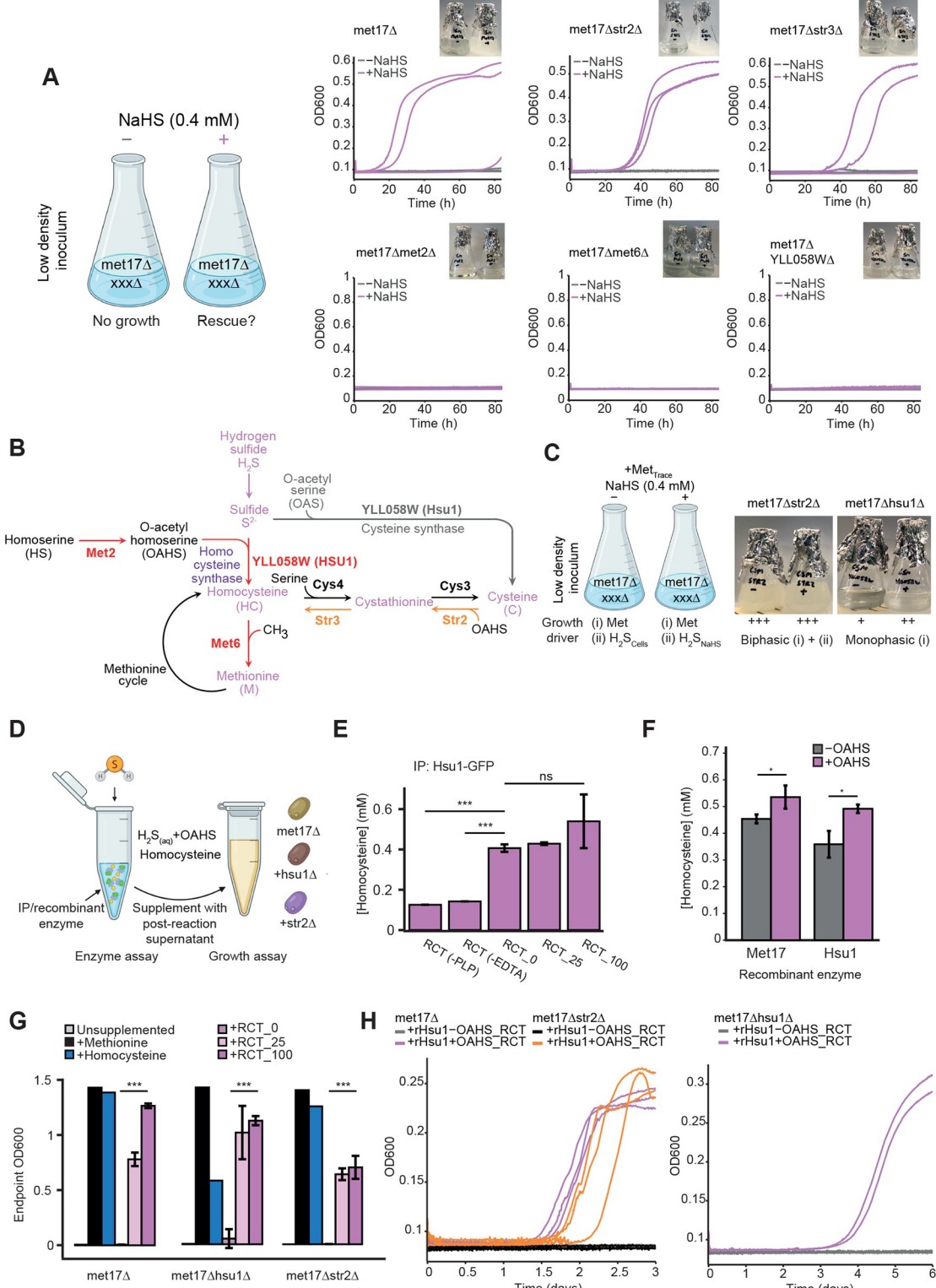

**Fig 3. Identification and characterization of YLL058Wp/Hsu1p as the homocysteine synthase responsible for hydrogen sulfide fixation.** (A) A targeted genetic screen of knockout strains lacking in addition to *met17Δ*, the genes encoding the homoserine (Met2p), methionine (Met6p) biosynthesis pathways, and selected orthologs of Met17p: Str3p, Str2p, and YLL058Wp. Cultures were inoculated at low density either in the presence or absence of NaHS, which rescues growth in *met17Δ* strains. Growth curves were recorded over 84 h with 3 biologically independent replicates for each strain (*n* = 3) as shown. A mutant strain

carrying *met17Δ* alone was used as control. Complete or partial loss of NaHS rescue phenotype in these cultures indicate involvement of the deleted enzyme in the metabolic bypass (purple). Images are representative of the endpoint following growth curve capture. (**B**) Simplified pathway that summarizes the deletion mutants with a no growth (red) or delayed growth (orange) phenotype during NaHS-mediated growth rescue. Metabolites bearing sulfur are marked in purple. Gray pathway indicates the possible conversion from OAS to cysteine that we observe in vitro with YLL058Wp. (**C**) Methionine limitation screen comparing *met17Δstr2Δ* and *met17ΔYLL058WΔ*. Low amounts of methionine were provided to fuel the first phase of growth, allowing cells to reach a critical density at which sufficient hydrogen sulfide leaked from cells to allow a second phase of growth. In this setup, NaHS supplementation enables the second phase of growth to proceed only if the bypass is functional by allowing cells to freely use $H_2S$ as a sulfur source regardless of limited concentrations of methionine. (**D**) Schematic indicating the setup of in vitro enzymatic reactions and growth assays used to assess the ability of OAHS reaction supernatants to rescue the indicated deletion strains (*met17Δ*, *met17ΔYLL058WΔ*, *met17Δstr2Δ*). Enzyme was sourced both from GFP immunoprecipitation (Hsu1-GFP) and recombinant (His-Hsu1, His-Str2, His-Met17) sources. (**E**) Quantification of homocysteine via Ellman's reagent and spectrometry using postreaction supernatants where immunoprecipitated Hsu1p-GFP was incubated with OAHS as the substrate. RCT_0, 25, and 100 indicate bead volumes of enzyme–resin slurry used in the reaction. RCT(-PLP or -EDTA) indicates reactions whereby pyridoxal 5′-phosphate (PLP) or EDTA has been omitted from the reaction buffer. Data are mean thiol concentrations ± SD where $n = 3$ biologically independent replicates. Student $t$ test was used to test for significance between indicated samples (***, $p < 0.0001$, ns = no significance). (**F**) Enzyme assay as in (E) utilizing recombinant enzymes purified by Ni-NTA affinity chromatography. Data are mean thiol concentrations ± SD where $n = 3$ biologically independent replicates. Student $t$ test was used to test for significance between indicated samples (*, $p < 0.05$, ns = no significance). (**G**) Growth assays whereby each strain was inoculated at low density and supplemented with 50 μl of the indicated in vitro reaction product ($n = 3$). Final $OD_{600nm}$ measured following 5 days of outgrowth. For OAHS reaction supplements, data are mean $OD_{600nm}$ ±SD where $n = 3$ biologically independent replicates. $OD_{600nm}$ reached in minimal media and supplementation with homocysteine or methionine ($n = 1$) are shown for comparison. (**H**) Growth assay as in (G) using postreaction supernatant generated from recombinant Hsu1 (rHsu1). Growth curves were captured for *met17Δ* and *met17Δstr2Δ* over 3 days, and for *met17Δhsu1Δ* over 5 days with 3 biologically independent replicates for each strain ($n = 3$). Schemes for Fig 3A, 3C, and 3D were created through BioRender.com. The data underlying this figure can be found in S1 Data. EDTA, ethylenediaminetetraacetic acid; NaHS, sodium hydrosulfide; Ni-NTA, nickel-NTA; OAHS, O-acetylhomoserine; OAS, O-acetylserine.

other candidate YLL058Wp have not previously been experimentally characterized, although based on its close phylogenetic relationship to Str2p, we hypothesized that it could possess cystathionine-γ-synthase activity or at least interact with similar substrates (S2E Fig).

To characterize how YLL058Wp and Str2p contributed to the bypass reaction, we devised an experiment in which we supplemented only a limited concentration of methionine into minimal media that was rapidly depleted (Fig 3C). In essence, the assay separates growth into two phases; the first phase is driven by methionine until the cells have reached the critical cell density that is required by the cells to overcome the *MET17* deficiency using the bypass enzyme. The *met17Δstr2Δ* strain maintained robust growth, both in the presence and absence of supplemented NaHS, indicating that the bypass reaction does not require Str2p. In contrast, growth of the *met17ΔYLL058WΔ* strain slowed after the initial biomass accumulation phase that is driven by the presence of methionine and did not achieve robust growth even in the presence of ample NaHS (Fig 3C). Thus, YLL058Wp is necessary for the cell to assimilate sulfide and sufficient to catalyze homocysteine biosynthesis from OAHS and sulfide, while Str2p appears to indirectly affect the bypass reaction. Considering the $H_2S$ dependency of YLL058Wp, we henceforth refer to it as **H**ydrogen **S**ulfide **U**tilizing-1 (Hsu1p).

## Hsu1p is a homocysteine and cysteine synthase that fixes sulfur from hydrogen sulfide

Having identified an enzyme required for the bypass reaction, we next sought to further characterize the levels and activity of Hsu1p, using a yeast strain that expresses a Hsu1-GFP fusion protein in an auxotrophic background lacking Met17p (*leu2Δura3Δmet17Δ*) [36]. We first verified by PCR the identity of the strain isolated from this library (S3A Fig). Monitoring GFP expression via flow cytometry, we observed that the GFP fluorescence increased 1.5-fold in minimal media supplemented with NaHS, indicating that *HSU1* expression is responsive to sulfide levels (S3B Fig). We next immunoprecipitated Hsu1-GFP from yeast cells grown under these conditions. Then, we tested in vitro if the immunopurified protein was capable of

transferring the sulfur from $H_2S$ onto OAHS to form homocysteine (Figs 3D and S3C). We also tested the ability of the enzyme to function as a cysteine synthase using O-acetylserine (OAS) as an alternative substrate, since this could potentially explain the growth of the *met17Δcys3Δ* strain in the presence of NaHS (Figs 3B and S3D). We quantified the formation of homocysteine or cysteine from OAHS or OAS, respectively, upon covalent linkage of inorganic sulfide to organic backbones using Ellman's reagent and spectrophotometry [37]. While the detection of homocysteine was hampered by the presence of a high background, the reaction of OAS to cysteine could be followed without strong interference. Cysteine synthase activity was observed at high enzyme concentrations (RCT_100; S3D Fig). To further validate homocysteine synthase activity in Hsu1p and to overcome the issue of high background, we generated recombinant versions of Hsu1p and Met17p by expressing the proteins in *E. coli* and purifying them via affinity chromatography (nickel-NTA (Ni-NTA)) and molecular weight cutoff (MWCO) filtration. We again tested for homocysteine synthase activity, this time using the purified recombinant enzymes in the presence or absence of OAHS substrate and reduced levels of PLP and ethylenediaminetetraacetic acid (EDTA) (Figs 3F and S3F). We observed an increase in homocysteine in the presence of OAHS for both Met17p and Hsu1p, again indicating that Hsu1p possesses homocysteine synthase activity. However, high enzyme levels were also required, indicating that this might be a secondary catalytic activity. We therefore tested whether the increase in homocysteine levels that we observed was relevant to our phenotype and linked the enzyme assay to a growth assay. Specifically, we asked whether supplementation with postreaction supernatant could rescue the growth of strains that previously had a growth defect in the absence of methionine or NaHS supplementation (*met17Δ*, *met17Δhsu1Δ*, *met17Δstr2Δ*, and *met17Δcys3Δ*). Supernatants obtained from the in vitro OAHS and OAS reactions (RCT_0, RCT_25, and RCT_100) were supplemented into minimal media without methionine and inoculated with the above strains. At 5 days postinoculation, the endpoint $OD_{600nm}$ of the OAHS supplemented cultures were compared against cultures directly supplemented with 0.15 mM methionine and homocysteine. Critically, we observed that supplementation with postreaction supernatant generated in the presence of immunoprecipitated Hsu1-GFP restores growth in the mutant backgrounds, akin to direct supplementation with an organosulfur compound (Fig 3G). Conversely, the supernatants taken from both OAHS and OAS reactions did not support growth of any tested strain in the absence of the immunoprecipitated enzyme (RCT_0; Figs 3G and S3E). Similarly, growth profiles captured from cultures supplemented with the OAHS postreaction supernatant generated in the presence of recombinant Hsu1p could also rescue the growth of the deletion strains, converse to supernatants that did not contain OAHS (Fig 3H). Hence, the reaction products formed in the presence of Hsu1p were sufficient to rescue the growth defects of the organosulfur auxotrophs.

## Evidence that Hsu1p is part of a sulfur limitation response in yeast

One prominent feature of our proteomic analysis was that *met17Δ* cells up-regulate enzymes involved in sulfur assimilation and methionine salvage (Fig 1E and 1F). This suggested that *met17Δ* cells experience sulfur limitation, owing to the reduced methionine and cysteine concentrations that would stem from the loss of its homocysteine synthase activity. Furthermore, an increase in the levels of Hsu1p is also triggered by increased sulfide concentrations (S3B Fig), which suggests that Hsu1p is regulated by the extracellular sulfide concentration. We hence sought to understand the conditions under which Hsu1p levels are increased in the presence of Met17p, which may help identify ecological conditions in which Hsu1p is essential, helping to explain its conservation. Previous studies have also shown that *HSU1* is localized within a regulon-like gene cluster that is believed to be up-regulated for the utilization of

alternative sulfur sources [38,39]. We therefore hypothesized that Hsu1p could function under sulfur limitation conditions in conjunction with other enzymes that enable the usage of alternative sulfur sources, such as $H_2S$. To explore this possibility, we analyzed the transcriptomic data acquired from a prototrophic yeast strain grown under sulfur limitation [40] together with genes identified as part of alternative sulfur usage regulons [39]. Comparing the genes within the datasets and the overlap between them revealed the up-regulation of sulfur assimilation enzymes and alternative organosulfur transporters/permeases upon sulfur limitation (Fig 4A). Hsu1p was found up-regulated in both datasets, together with Mmp1p (high-affinity S-methylmethionine permease), Jlp1p (Fe(II)-dependent sulfonate/alpha-ketoglutarate dioxygenase), Yct1p (high-affinity cysteine-specific transporter), Soa1p (sulfonate and inorganic sulfur transporter), and Bds1 (bacterially derived sulfatase). Therefore, the induction of *HSU1* in parallel with other genes involved in the uptake or utilization of alternative sulfur sources suggested a functional association of these genes to a sulfur limitation response. Furthermore, the up-regulation of sulfur assimilation enzymes (Met1p, Met3p, Met8p, Met10p, Met16p, and Sul2p) is similarly observed in the *met17Δ* proteomes (Fig 1G), which is strongly suggestive of a common sulfur limitation signature.

To experimentally test this hypothesis, we modified our minimal medium, replacing ammonium sulfate with ammonium chloride, and titrating magnesium sulfate ($MgSO_4$, 0 to 500 mg/L) together with magnesium chloride ($MgCl_2$, 500 to 0 mg/L)) to maintain the magnesium concentration while modulating the sulfate concentration (see Materials and methods). In a background of *MET17* prototrophy, the Hsu1-GFP strain demonstrated an increase in GFP fluorescence during conditions of sulfur limitation (Fig 4B; [$MgSO_4$] = 5 mg/L). This phenotype is abrogated when sulfate concentrations are increased (SM, or [$MgSO_4$] = 50 or 500 mg/L), suggesting that the increase in Hsu1p levels is conditional upon and regulated by either sulfur or organosulfur limitation. Growth profiling of both the *HSU1* and *hsu1Δ* strains during sulfur limitation reveals that the presence of Hsu1p confers a moderate growth advantage under these conditions (Fig 4C). To further investigate this observation, we subjected both strains to a competition assay against a control strain that expresses *HSU1*. In these experiments, our experimental *HSU1* strain carried a Kan4MX marker (Kan4MX::*his3Δ*) and a pHLUM plasmid expressing *HIS3*, *LEU2*, *URA3*, and *MET17*. The control strain carried *HSU1* and had no Kan4MX marker, allowing us to use kanamycin as a marker to distinguish the experimental *HSU1* and *hsu1Δ* strains from the control *HSU1* strain. Following 48 h of competition in both sulfur-replete (SM) and sulfur-limited (SL_5) conditions, we plated a sample of the cultures onto minimal media, followed by replica plating onto SM or SM+G418 to assess the proportion of Kan+ cells in the original culture (Fig 4D). In agreement with the growth curve analysis (Fig 4C), we observed that *HSU1* expression confers a moderate but significant 1.2-fold growth advantage under sulfur-limited conditions, which was not observed when sulfur was replete (Fig 4E). Conversely, *HSU1* expression appears to be detrimental in sulfur-replete conditions, as the *hsu1Δ* strain had a similar growth advantage. This suggests that cells under rich conditions suppress *HSU1* expression to alleviate this growth defect. We suspect this growth defect may be a result of substrate competition, most likely for OAHS, which is shared among Met17p, Str2p, and Hsu1p. Therefore, it appears that *HSU1* expression is tightly regulated, is strongly conditional on changes in sulfide concentration, and its presence may provide a competitive advantage to cells growing in low-sulfur conditions.

## Discussion

Sulfur assimilation is a fundamental process of life, yet the biological response to sulfur limitation is still not fully understood. This study was stimulated by an aim to identify the

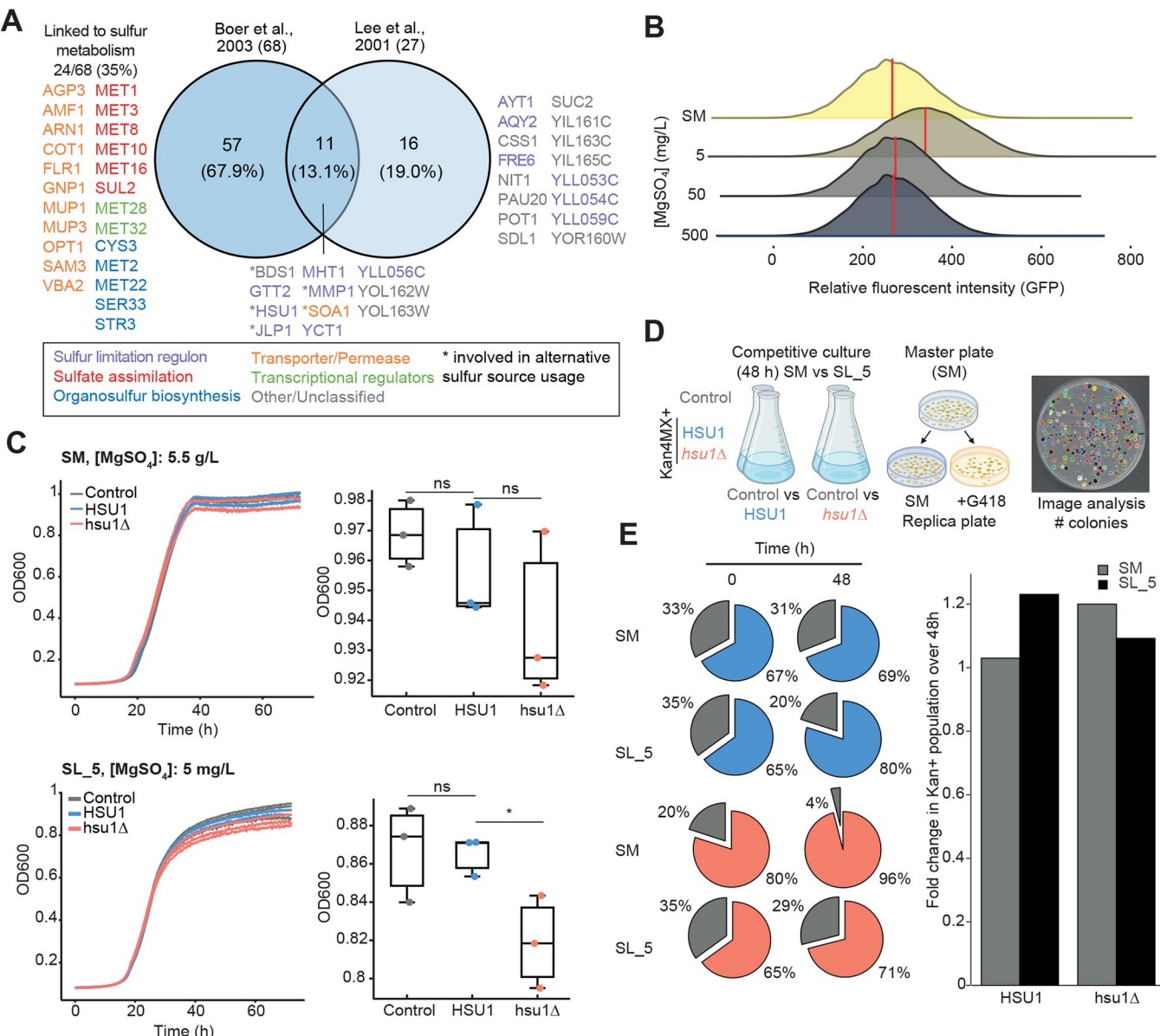

**Fig 4. Hsu1p confers a growth advantage under prolonged sulfur limitation in the presence of Met17p. (A)** Comparison of genes identified by transcriptomics that are up-regulated under sulfur limitation against genes identified as part of the sulfur limitation regulon regulated by Met4p [39,40]. Of the 68 genes up-regulated in response to sulfur limitation, 24 (35%) have functional annotations related to sulfur metabolism or transport. Gene functions are classified as indicated. * indicates gene products that have a defined enzymatic function related to the usage of alternative sulfur sources. **(B)** Profiling of Hsu1-GFP by flow cytometry in a prototrophic strain exposed to sulfur limitation conditions via titration of MgSO$_4$ (mg/L). SM indicates minimal media containing standard concentrations of (NH$_4$)$_2$SO$_4$ (5 g/L) and MgSO$_4$ (500 mg/L) giving a total sulfate concentration of 5.5 g/L. Histograms indicate population of GFP+ cells measured (>20,000); red line indicates relative median fluorescence intensity of the captured population. **(C)** Growth profiles and endpoint OD$_{600nm}$ of control, *HSU1*, and *hsu1Δ* strains under sulfur-replete (SM) and limited (SL_5) conditions. All strains express MET17 due to the repair of auxotrophies via plasmid complementation with pHLUM. Opposed to the *HSU1* (*his3Δ*) and *hsu1Δ* strains, the control strain does not harbor any gene knockout and is Kan−. The *HSU1* strain is *his3Δ*, which serves to control for the position effect of the Kan4MX cassette, thereby making it comparable to *hsu1Δ*. **(D)** Setup of a growth competition assay competing the control strain with either *HSU1* or *hsu1Δ* strains grown in SM vs. SL_5. Since both knockout strains harbor a Kan4MX cassette their frequency can be assessed by replica plating onto SM±G418. The number of positive colonies can be counted via image analysis and presented as a ratio between the two replica plates per condition and per time point. **(E)** Frequency of Kan+ and Kan− populations across all conditions and time points tested (pie charts). Adjacent barplot summarizes the fold change in Kan+ populations over 48 h between SM and SL_5 conditions. The data underlying this figure can be found in S1 Data.

biochemical basis for a paradoxical yeast growth phenotype reported by Cost and Boeke [19,21] when they established *MET17* as an auxotrophic marker for yeast genetic experiments. Certain specific conditions render this auxotrophic marker "leaky," whereby cells are somehow able to overcome an otherwise strict organosulfur auxotrophy when inoculated at high cell density. They also identified the overproduction and leakage of sulfide ions, which we have termed sulfide overflow, as a peculiarity of *met17Δ* cells. In searching for the biochemical basis that unites these observations, we find evidence for a previously overlooked sulfur fixation route. This protein, herein named Hsu1p, allows cells to reincorporate the sulfide ions that accumulate to growth relevant concentrations in high-density cultures. Hsu1p resolves organosulfur auxotrophy by substituting for Met17p, catalyzing the incorporation of inorganic sulfide ions with OAHS to form homocysteine, thereby replenishing the central organosulfur pool and rescuing the *met17Δ* growth defect in a cell–cell cooperative manner. Our data also suggest an explanation for the temperature dependency of the rescue; higher temperatures drive the phase equilibrium towards formation of gaseous $H_2S$, which is readily lost from the system.

While this manuscript was under consideration, a parallel study was published as a preprint [41]. Although both studies use different approaches, they agree on the enzymatic function of Hsu1p and the involvement of $H_2S$ as a substrate.

Transcriptomic data recorded by Oliver and colleagues [38,40] show that Hsu1p is induced during sulfur limitation, and we have experimentally observed that the presence of Hsu1p and Met17p can confer a growth advantage under these conditions. However, how the cell regulates the activity of both homocysteine synthases to maintain growth without depleting their shared substrate pools remains to be studied, although sustained activity of both during replete conditions appears undesirable. We observed that the accumulation of $H_2S$ overcomes the loss of Met17p in the presence of Hsu1p, either directly provided or generated because of sulfide overflow in high-density cultures.

The ability of met17Δ mutants to overcome methionine auxotrophy with Hsu1p illustrates an interesting property of feedback control in metabolism. Negative feedback is a common feature of metabolic pathways that drives upstream enzymes to be up-regulated in an attempt to increase flux through the pathway when low concentrations of downstream metabolites are detected [42–44]. Feedback or endpoint inhibition has been shown to exist in various biosynthetic pathways in yeast including the synthesis of methionine, arginine, branched-chain amino acids, lysine, and aromatic amino acids [15,45,46]. We observe that in *met17Δ* mutants, the up-regulation of these upstream enzymes can lead to buildup of sufficient $H_2S$ to allow Hsu1p to bypass the metabolic deficiency with a reaction that does not occur in its absence. Therefore, gaseous $H_2S$ could theoretically function as a chemical messenger, a shared metabolite that could allow cells within the culture to alter their behavior and possibly serve as a quorum sensing mechanism.

## Materials and methods

### Yeast culture

Yeast strains (S1 Table) were streaked from cryostock onto YPD, synthetic minimal medium (SM, 6.8 g/L YNB without amino acids (Sigma, Y0626; Lot: MKCF2863), 2% w/v glucose (Sigma), 2% w/v agarose (Life Technologies)) or supplemented synthetic minimal medium (SM+histidine (H, 20 mg/L), leucine (L, 100 mg/L), uracil (U, 20 mg/L), and methionine (M, 20 mg/L) and incubated for 3 days at 30˚C before a single colony was picked to inoculate 20 ml of media in a conical flask and subsequently incubated at 30˚C for growth under various conditions as outlined in the experiments above. Prototrophic (WT) and SeMeCo strains were revived onto SM medium as described in ref [28], while strains harboring only *met17Δ* were streaked onto SM+methionine. Strains where auxotrophy complementation was introduced

were cultured in selective media (variations of SM±HLUM) at all times unless otherwise stated. Cultures that required activation of the bypass reaction (high-density inoculations or supplementation with NaHS) were cultured at 25˚C to maximize growth (see Fig 1B).

## Growth curve profiling

For the growth analysis of amino acid auxotrophs (Fig 1C), deletion mutants from the BY4741 prototrophic collection and the *met17Δ* strain were revived on SM or SM+Methionine for 36 h. Single colonies were picked for each strain, resuspended in water (Millipore, autoclaved) in a short 96-well plate, and then pinned onto synthetic complete (SC) agar medium using the Singer RotorHD. After 72 h, colonies were used to inoculate a 96-well plate containing SM liquid medium and repinned onto SM solid medium. Growth curves were recorded by placing the plate containing liquid medium in a Tecan Infinite M200 PRO spectrophotometer to determine optical density at 600 nm (10 flashes/read) for 72 h. The solid media plate was incubated at 30˚C for 72 h. Growth curve analysis was conducted using the growfit package in R (v4.0.2) and $\Delta OD_{600nm}$ was calculated, extracted, and plotted using custom scripts. In NaHS supplementation and the other growth curves, growth curve profiling was extended to 72 to 100 h to capture a complete profile and plotted directly using custom scripts.

## Systematic growth analysis of *met17Δ* in minimal media

Single colonies of the *met17Δ* and the prototrophic WT strain ($n = 3$ replicates per genotype) were inoculated in SM+Methionine, grown overnight at 30˚C in Erlenmeyer flasks, diluted 1:20 in fresh SM+Methionine or SM media, and incubated for 4 h. Cells were collected, washed in autoclaved $H_2O$ (Millipore), resuspended in SM, and the $OD_{600nm}$ of each strain was adjusted to 0.6. Each set of replicates ($n = 3$ for the deletion strain, and $n = 3$ for the control strain) were then transferred in a deep 96-well plate where a sequential 1:1 dilution of 100 μl culture was performed with 100 μl SM media to obtain an $OD_{600nm}$ range of 0.3 to approximately 0.0003. $4 \times 200$ μl of each well was transferred to $4 \times 96$-well microtiter plates (Costar). The plates were sealed using a nonpermeable, clear foil (Roth) and were incubated at 22˚C, 26˚C, 30˚C, and 37˚C without shaking. Optical density was recorded by placing the plates in an Infinite M200 Pro plate reader (Tecan), and $OD_{600nm}$ was obtained from the median of 9 multi-well reads per well. Subsequently, for each plate, the median $OD_{600nm}$ of all blank ($n = 8$ per plate) values was subtracted from obtained $OD_{600nm}$ values.

## Hydrogen sulfide production quantification

Qualitative assessment of hydrogen sulfide production was indicated by the precipitation of lead sulfate in growth media containing 0.1 g/L lead(II) acetate. Quantification of hydrogen sulfide was conducted using AzMC, which undergoes selective reduction of the azide moiety in the presence of hydrogen sulfide to form AMC. In brief, 10 μl of filtered yeast media was mixed with 200 μl of 10 μM AzMC and incubated at RT for 30 min before the fluorescence was recorded on a Tecan spectrophotometer at $_{em}\lambda = 450$ nm ($_{ex}\lambda = 365$ nm). For the generation of hydrogen sulfide, we utilized NaHS as the donor, reconstituted with water to a stock concentration of 20 mM and a working concentration in the medium of 0.2 or 0.4 mM.

## Metabolite quantification and isotopic tracing by LC–MS/MS

**Sample preparation.** For intracellular and extracellular metabolite quantification, yeast cells were cultured in synthetic minimal medium and metabolite extraction and quantification was performed as described previously [25,28]. Purity of YNB was assessed by direct

measurement of prepared liquid media. For isotope tracing experiments, yeast samples were essentially treated and acquired as above. Briefly, single colonies were resuspended in sterilized $H_2O$ and $OD_{600nm}$ was determined. For each biological replicate, $2 \times 1.1$ ml of 1.7 g/l YNB without ammonium sulfate and amino acids (Sigma), 2% glucose, and naturally labeled ammonium sulfate (5 g/l) or ammonium sulfate-$^{34}S$ (5 g/L, Sigma, $> = 98\%$ $^{34}S$) as situated in $2 \times 1.5$ ml microcentrifuge tubes were inoculated to an $OD_{600nm}$ of 0.03. Cells were incubated in closed vials at 30°C without shaking for 48 h and harvested by centrifugation (3 min, 3,220*g*). The supernatant was taken off and passed through a syringe filter (0.22 μm). Corresponding pellets were pooled and extracted in 100 μl prewarmed ethanol, vortexed, incubated for 2 min in a water bath at 80°C, vortexed, and transferred again for 2 min to a water bath at 80°C. Extract was cleared by centrifugation (15 min, 16,000*g* at 4°C) and the supernatant was transferred to HPLC glass vials for analysis. The filtered supernatant of each tube (0.9 ml) was reconstituted by addition of fresh media constituents as supplied by addition of 50 μl glucose (40% v/v), and 100 μl YNB w/o ammonium sulfate and amino acids (17 g/L) and respective ammonium sulfate (50 g/L). Media was then inoculated to an $OD_{600nm}$ of 0.003, grown for 48 h, and cells were harvested and extracted as described above.

**Data acquisition.** Data was acquired by LC–MS/MS using a high-pressure liquid chromatography system (1290 Infinity, Agilent Technologies) coupled to a Triple Quad mass spectrometer (6470, Agilent Technologies). Samples were separated using an Acquity UPLC BEH Amide 1.7 um, $2.1 \times 100$ mm column. The gradient consisted of mobile phase A (water with 100 mM ammonium carbonate) ramped against mobile phase B (acetonitrile) with a flow rate of 0.3 ml as follows: 0 min, 70% B, 3 min, 70% B, 7 min, 40% B, 8 min, 40% B, 8.1 min, 70% B, followed by 1.9 min equilibration. Mass spectrometer was operating in MRM mode with the following source settings: gas temperature 200 C, Gas flow 13 L/min, nebulizer 60 psi, sheath gas temp 300°C, sheath gas flow 11 L/min, Capillary Voltage 3500 (positive mode) or 2500 (negative mode), and a ΔEMV of 100 (both modes). Injection volume was 1 μl.

Metabolites were identified by matching retention time and transitions as confirmed by analytical standards. Homoserine, homocysteine and methionine were analyzed in positive mode using the transitions homoserine: 120.1 to 44.0; homocysteine: 136.0 to 56.0 and methionine 150.1 to 132.9 (unlabeled) or 152.1 to 134.9 ($^{34}S$ labeled). Reduced glutathione was analyzed in negative mode with the transitions 306 to 143 (unlabeled) and 308 to 143 ($^{34}S$ labeled).

## Ortholog mapping and phylogenetic analyses

Orthologs of Met17p were identified in the Eggnog Database (http://eggnog5.embl.de/; [32]) using Met17p as a query protein and considering orthologs from the Eukaryota taxa. This yielded 753 proteins from the KOG0053 orthogroup, from which we only considered orthologous proteins belonging to *S. cerevisiae* (7 proteins; S2 Table). All these genes were considered for our screen of H2S utilization in a *met17Δ* background.

To understand the phylogenetic relationship between the *S. cerevisiae* orthologs of Met17p (S2E Fig), each of these sequences was then used as a query in NCBI pBLAST (https://blast.ncbi.nlm.nih.gov/) focusing on the genomes of 26 model organism species including 17 fungal species, 2 plant species, 5 animal species, a choanoflagellate (*Salpingoeca rosetta*), and *E. coli* (S4 Table). Duplicates and highly similar proteins were removed (Similarity score over 60 using the globalms algorithm from biopython's pairwise2 package with match_points = 1, mismatch_points = −1, gap_open = −0.5, gap_extension = −0.1 and penalize_end_gaps = False) [47]. Multiple Sequence Alignments (MSAs) were then generated using mafft v7 with–genafpair and–maxiterate 1000 options. The MSAs were trimmed using TrimAl 1.2 with -gappyout [67] and trees were generated from the trimmed alignment using IQTREE v1.6.12 and the

model LG+R5 with *E. coli* protein EEV6508859.1 as the outgroup [68]. The full tree is included in the Supporting information, and the topology of this tree (but with branch lengths not drawn to scale) was used to construct the summary tree manually (S2E Fig).

### Immunoprecipitation of Hsu1-GFP

The strain was streaked from the GFP library and the coding region was verified via specific primers [20] (S3 Table). To maximize the recovery of Hsu1-GFP protein, the strain was cultured for 5 days in 150 ml minimal media supplemented with leucine, uracil, and 0.4 mM NaHS at 25˚C. Following this, cells were pelleted by centrifugation at 3,220*g* for 5 min at 4˚C and resuspended in an appropriate volume of Y-PER Yeast Protein Extraction Reagent (Thermo Fisher) containing 1× Protease Inhibitor cocktail (P8215, Sigma). Purified protein extract was obtained following the manufacturer's protocol, to which 250 μl of GFP-clamp resin (in-house, Crick-Structural Biology Scientific Technology Platform) was added to the lysis solution and incubated at 4˚C for 2 h with rotation.

### Generation of recombinant Hsu1p and Met17p

Genomic DNA was isolated from a prototrophic BY4741 strain [28] following an established protocol [48]. The coding sequence from each protein was cloned via two rounds of PCR, firstly to amplify the coding sequence and secondly to attach the leader sequences required for subcloning into a bacterial expression vector via ligation independent cloning (LIC; primers are available in S3 Table). The His-tagging, thioredoxin-fusion expression vector, pN-TrxT, was obtained from Addgene (#26101; [49]) and digested with *BsaI* to release the *sacB* stuffer fragment and linearize the vector. Following gel purification of both linearized vector and PCR fragments, LIC was initiated following treatment with T4 DNA polymerase in the presence of either dGTP (vector) or dCTP (inserts). Vector and inserts were mixed at a 1:3 ratio and incubated at RT for 5 min to facilitate annealing. Nicks were sealed upon transformation into competent DH5α *E. coli* (NEB) and plated onto selective media (LB+10% sucrose), which eliminates clones carrying undigested vector. Clones were then picked and insertion was verified by PCR and sequencing before amplification and extraction of the plasmids via miniprep (Qiagen). Plasmids were then retransformed into competent BL21 *E. coli* (NEB) and selected on LB+Kan plates. For recombinant protein production, a single colony was picked for each recombinant protein and used to inoculate the initial 1L culture, to which IPTG was added at a final concentration of 400 μM when $OD_{600nm}$ = 0.8. Cultures were then grown overnight at 22˚C with shaking and harvested the following morning via centrifugation at 4,000 rpm for 10 min. Bacterial pellets were lysed in B-PER Bacterial Protein Extraction Reagent (Thermo) according to the manufacturer's instructions. Presence of His-tagged protein was initially confirmed using His-Tag Protein Expression Check Kit (Abcam), before lysates were subjected to two rounds of Ni-NTA-based affinity purification (Thermo) and one round of MWCO centrifugation (Merck) following the manufacturer's protocol. Following purification, protein solution was dialyzed into 20 mM Tris–HCl buffer (pH 7.8), and protein was quantified via BCA assay (Thermo). Protein gels (Invitrogen) were run and stained with Instablue Coomassie Blue (Abcam) to verify purified protein was of the expected size. An equal amount of each enzyme was then added to the reaction tubes to which 200 μM PLP, 1 mM EDTA, and 5 mM OAHS (Insight Biotech) were added and initiated as below.

### Hydrogen sulfide utilization enzyme and growth assays

For immunoprecipitated Hsu1-GFP, beads were washed with twice with Y-PER Yeast Protein Extraction Reagent (Thermo Fisher), followed by a final wash with reaction buffer (0.1 M

potassium phosphate buffer (pH 7.8), 1 mM EDTA, 200 µM pyridoxal 5′-phosphate (PLP), all Sigma). Enzyme-bead complexes were then resuspended in 100 µl of reaction buffer containing 5 mM OAS (Sigma) or OAHS and placed with cap open into Eppendorf tube holders contained in a large beaker. A smaller beaker containing 20 mM NaHS solution was also placed into the large beaker that was sealed in parafilm and made airtight. The entire setup was placed at 30°C and incubated for 24 h, during which $H_2S$ would outgas from the NaHS solution, dissolve, and react with OAS/OAHS in the presence of the enzyme (S3C Fig). The setup was identical for reaction tubes containing recombinant enzymes opposed to enzyme-bead complexes, with the exception of EDTA and PLP, whose concentration was reduced by half to reduce background. Tubes were then retrieved from the beaker and placed in an Eppendorf Thermomixer and shaken for 2 h at RT to remove residual $H_2S$ before quantification via Ellman's reagent (Thermo Fisher). For growth assays, 50 µl of reaction supernatant was transferred to 15 ml round bottom tubes containing 3 ml of media and incubated with shaking for up to 5 days at 25°C before endpoint $OD_{600nm}$ was measured using a cuvette spectrophotometer.

## Quantification of cysteine/homocysteine

Quantification of cysteine/homocysteine resulting from the OAS/OAHS reaction, respectively, was performed using Ellman's reagent (5,5-dithio-bis-(2-nitrobenzoic acid, ThermoFisher; [37]), which preferentially reacts with sulfhydryl groups to yield a colored product. In brief, a standard curve using cysteine (0 to 0.2 mM, Sigma) was generated and incubated with the reagent following the manufacturer's protocol at a 1:1 ratio. Absorbance was measured on a Tecan Infinite M200 PRO spectrophotometer at $\lambda = 412$ nm, and concentration was calculated using the standard curve.

## Sulfur limitation assays

**Establishing sulfur-limited growth conditions.** To create sulfur limitation media, a custom formulation of YNB was first generated from stock solutions of each of the components, in which ammonium sulfate and magnesium sulfate, the two largest sources of sulfate at 5 g/L and 0.5 g/L, respectively, were omitted. Then, ammonium chloride was added to replace the nitrogen source without addition of sulfate, while magnesium chloride was added to replace the magnesium source, titrated with magnesium sulfate, giving a final sulfate concentration of 5 mg/L, a 1,000-fold decrease in overall sulfate. A sulfur-replete version of the media was generated by simply adding ammonium sulfate and magnesium sulfate back to the custom SM to give a final sulfate concentration of 5.5 g/L. This was used in the subsequent growth curve and competition experiments to ensure that deviations observed are not due to the use of custom media.

**Competition assay.** The control strain (pHLUM; [50]) and knockout harboring strains that contain the Kan4MX cassette from the prototrophic knockout library (*HSU1* and *hsu1Δ*) were revived onto SM following standard procedures. It is important to note that all three strains contain the pHLUM that complements the original auxotrophies (*his3Δ1*, *leu2Δ*, *ura3Δ*, *met17Δ*); consequently, the *his3Δ* deletion in the *HSU1* strain is phenotypically compensated for by pHLUM, while still retaining the Kan4MX marker and therefore controls for the positional effects of the cassette when directly compared against *hsu1Δ*. The Kan4MX could therefore also be used to distinguish the control and knockout strains during competitive culture. For the setup of the competition culture, a preculture of each strain was established in SM, after which cells were washed thrice with autoclaved $H_2O$ (Millipore) to remove residual sulfate. The knockout strains were then separately inoculated into SM (sulfur replete) or SL_5 (sulfur limited) media and competed against an equal amount of control strain based

on $OD_{600nm}$ with a combined $OD_{600nm}$ no greater than 0.05. Cultures were then incubated at 25˚C to maximize the activity of Hsu1p for 48 h, after which a 10-μl sample of each culture was plated onto SM solid medium. Plates were then incubated for 72 h, before being replica plated onto SM or SM+G418 agar. After 24 h, plates were imaged and colonies were counted using ImageJ.

## Supporting information

**S1 Table. Source and description of yeast strains used in this study.**
(XLSX)

**S2 Table. Met17 orthologs in *S. cerevisiae*.**
(XLSX)

**S3 Table. Primers used in this study.**
(XLSX)

**S4 Table. List of model organisms used for phylogenetic analysis.**
(XLSX)

**S1 Data. Raw data files for all figures generated in this study.**
(RAR)

**S1 Fig. Characterization of the *met17Δ* prototrophic growth phenotype in the absence of methionine. (A)** Streak cultures of prototrophic WT and *met17Δ* strains on agar with and without methionine, imaged at 3 and 8 days incubation at 30˚C. **(B)** LC–MS analysis of the YNB component of our minimal media. Quantified metabolites were compared to their respective standards and FC calculated. **(C)** Liquid cultures of *met17Δ* strain in minimal media without methionine inoculated at 6 different cell densities (0.01 to 0.3) and cultured at 4 different temperatures (22, 26, 30, 37˚C) for 24 h. Gray indicates WT control strain that is prototrophic for methionine cultured at 30˚C. Lines indicate three replicates per condition, where $n =$ 3. **(D)** Growth of *met17Δ* strain in 20 ml cultures. Starting $OD_{600nm}$ of cultures varied between 0.01 and 0.07 and growth was assessed after 48 h of culture in minimal media without methionine. **(E)** Quantification of methionine auxotrophs in exponentially growing SeMeCo cultures. Data are mean ± SEM from 3 biologically independent replicates ($n = 3$). * indicates $p < 0.05$ via Student *t* test. Growth curves captured from indicated amino acid auxotrophic strains compared against the prototrophic and *met17Δ* strains. Inserts depict colony growth following 72 h of culture. The data underlying this figure can be found in S1 Data. FC, fold-change; LC–MS, liquid chromatography mass spectrometry; SeMeCo, self-establishing metabolically cooperating community; WT, wild-type; YNB, yeast nitrogen broth.
(TIF)

**S2 Fig. Cell density dependency and involvement of sulfur assimilation mutants in the utilization of $H_2S$. (A)** Representative mass spectroscopy traces of methionine, reduced glutathione in cultures supplemented with $^{32}S$ and $^{34}S$-labeled ammonium sulfate. Gray and yellow traces indicate $S^{32}$ and $S^{34}$ versions of the metabolite. **(B)** Schematic depicting the sulfur assimilation pathway from sulfate to sulfide in yeast. Metabolites carrying the sulfur are indicated in light purple. **(C)** Growth screens for loss of $H_2S$ utilization in sulfur assimilation deletion mutants upstream of *MET17* mapped against the pathway in **(B)**. For each strain, growth curves were captured from 3 biologically independent cultures over 84 h. **(D)** Screen for loss of $H_2S$ utilization in strains lacking Irc7p, YML082Wp, and YHR112Cp. For each strain, growth curves were captured from 3 biologically independent cultures over 84 h. All mutants tested

are in a background of *met17Δ*. Scheme for Fig SB was created with BioRender.com. (**E**) Tree illustrating the phylogenetic relationships between orthologs of Met17p found in *S. cerevisiae*. After identifying orthologs in *S. cerevisiae*, orthologs in select model organisms were identified using BLAST and a multiple sequence alignment and tree were generated (see Materials and methods). The full tree is available in the Supporting information (S1 Data) and this summarizes the tree structure. Bold outlines indicate the *S. cerevisiae* proteins and other nodes represent groups of orthologs from the indicated clades. The major reaction catalyzed by each enzyme is annotated alongside as per the Uniprot/Rhea-annotated reactions database. The data underlying this figure can be found in S1 Data.
(TIF)

**S3 Fig. Characterization of Hsu1-GFP strain and enzymatic activities. (A)** PCR verification of the Hsu1-GFP ORF using genomic DNA isolated from the Hsu1-GFP strain using primers within the HSU1 and GFP open reading frames and the ADH1 terminator. All products were of the expected size as indicated by gel electrophoresis. (**B**) Flow cytometry analysis and confirmation of GFP intensity in response to histidine and methionine deficiency with $H_2S$ supplementation. Numbers indicate median fluorescence intensity, from approximately 40,000 events captured per strain. The *met17Δ* strain was used as a negative, nonfluorescent control, while the Met6p-GFP strain was used as a positive, highly fluorescent control. (**C**) Schematic of $H_2S$ utilization enzyme assay. Off-gassing of $H_2S$ from a 20-mM NaHS solution dissolves into reaction tubes containing either OAHS or OAS as substrate. Insert indicates actual setup of reactions. (**D**) Quantification of thiol concentrations via Ellman's reagent using supernatant from enzyme assays where immunoprecipitated Hsu1-GFP was incubated with either OAS as the organic substrate. RCT_0, 25, and 100 indicate volumes of enzyme–resin slurry used in the reaction. Data are mean thiol concentrations ± SD where *n* = 3 biologically independent replicates. (**E**) Growth rescue assay of *met17Δcys3Δ* strain with the product of the OAS reaction. RCT_0, 25, and 100 indicate the volume of enzyme–resin complex used, which approximates increasing enzyme concentrations. $OD_{600nm}$ was measured from two independent sets of experiments as indicated. (**F**) Coomassie staining of recombinant Trx-His-tagged proteins purified via two rounds of nickel affinity chromatography. Molecular weight in kDa for the ladder and proteins are as indicated. Samples were run in triplicate in successive 1:2 dilutions. Scheme for (A) was generated with SnapGene software (www.snapgene.com). Scheme for (C) was created with BioRender.com. The data underlying this figure can be found in S1 Data. HSU1, Hydrogen Sulfide Utilizing-1; OAHS, O-acetylhomoserine; OAS, O-acetylserine.
(TIF)

## Acknowledgments

We thank all our lab members and Prof. Judith Berman for critical discussion and comments to the manuscript. We also thank Dr. Svend Kjaer and Dr. Phillip Walker at the Crick Structural Biology STP for their technical support.

## Author Contributions

**Conceptualization:** Jason S. L. Yu, Markus Ralser.

**Data curation:** Jason S. L. Yu.

**Formal analysis:** Jason S. L. Yu, Benjamin M. Heineike, Johannes Hartl, Simran K. Aulakh, Clara Correia-Melo, Andrea Lehmann, Oliver Lemke, Markus Ralser.

**Investigation:** Jason S. L. Yu, Benjamin M. Heineike, Johannes Hartl, Simran K. Aulakh, Clara Correia-Melo, Andrea Lehmann, Oliver Lemke, Federica Agostini, Cory T. Lee.

**Methodology:** Jason S. L. Yu, Johannes Hartl, Simran K. Aulakh, Clara Correia-Melo, Vadim Demichev, Christoph B. Messner, Michael Mülleder.

**Project administration:** Jason S. L. Yu, Markus Ralser.

**Resources:** Vadim Demichev, Christoph B. Messner, Michael Mülleder.

**Software:** Vadim Demichev, Christoph B. Messner, Michael Mülleder.

**Supervision:** Markus Ralser.

**Validation:** Jason S. L. Yu, Benjamin M. Heineike, Johannes Hartl.

**Visualization:** Jason S. L. Yu, Johannes Hartl, Simran K. Aulakh, Clara Correia-Melo, Federica Agostini, Cory T. Lee, Markus Ralser.

**Writing – original draft:** Jason S. L. Yu, Benjamin M. Heineike, Markus Ralser.

**Writing – review & editing:** Jason S. L. Yu, Benjamin M. Heineike, Johannes Hartl, Markus Ralser.

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
