## [Editor Report · Decision Letter 0]

17 Oct 2022

Dear Markus, 

Thank you for submitting your revised Review Commons manuscript entitled "Inorganic sulfur fixation via a new homocysteine synthase allows yeast cells to cooperatively compensate for methionine auxotrophy" for consideration as a Research Article by PLOS Biology.

Your manuscript has now been evaluated by the PLOS Biology editorial staff, as well as by an academic editor with relevant expertise, and I'm writing to let you know that we would like to send your submission out for re-review (I had hoped to avoid re-review, but the Academic Editor currently has too many commitments to check the responses and revisions themselves).

However, before we can send your manuscript back to reviewers, we need you to complete your submission by providing the metadata that is required for full assessment. To this end, please login to Editorial Manager where you will find the paper in the 'Submissions Needing Revisions' folder on your homepage. Please click 'Revise Submission' from the Action Links and complete all additional questions in the submission questionnaire.

Once your full submission is complete, your paper will undergo a series of checks in preparation for re-review. After your manuscript has passed the checks it will be sent out for review. To provide the metadata for your submission, please Login to Editorial Manager (https://www.editorialmanager.com/pbiology) within two working days, i.e. by Oct 19 2022 11:59PM.

Kind regards,

Roli

Roland Roberts, PhD

Senior Editor

PLOS Biology

rroberts@plos.org

---

## [Decision Letter · Decision Letter 1]

2 Nov 2022

Dear Markus,

Thank you for your patience while we considered your revised Review Commons manuscript "Inorganic sulfur fixation via a new homocysteine synthase allows yeast cells to cooperatively compensate for methionine auxotrophy" for publication as a Research Article at PLOS Biology. This revised version of your manuscript has been evaluated by the PLOS Biology editors, the Academic Editor, and one of the original reviewers.

Based on the review and on our Academic Editor's assessment of your revision, we are likely to accept this manuscript for publication, provided you satisfactorily address the following data and other policy-related requests.

IMPORTANT: Please attend to the following:

a) We are aware of the closely related paper from the Carvunis lab, whose preprint you cite in your paper. The Academic Editor was concerned to minimise confusion in the field around nomenclature and wondered if you could possibly liaise with Carvunis and colleagues to try to ensure harmonisation over "Hsu1"/YLL058W (I see that their preprint sticks with the latter, but this may have change since initial deposition).

b) Still on the topic of the Carvunis study, I see that it's listed as ref 41 in your bibliography, but there's no in-text citation to it; please could you rectify this? I couldn't find a citation for ref 40, either. If you do contact Carvunis, it may be helpful to obtain a more up-to-date citation ("in press" at a specific journal, rather than the preprint); if it publishes soon, you could also update your paper at the proof stage.

c) Please provide a blurb, according to the instructions in the submission form.

d) Please address my Data Policy requests below; specifically, we need you to supply the numerical values underlying Figs 1BCDEF, 2E, 3AEFGH, 4BCE, S1BCEF, S2ACD, S3BDE, either as a supplementary data file(s) or as a permanent DOI’d deposition.

e) Please cite the location of the data clearly in all relevant main and supplementary Figure legends, e.g. “The data underlying this Figure can be found in S1 Data” or “The data underlying this Figure can be found in https://doi.org/XXXX”

We expect to receive your revised manuscript within two weeks. 

*Published Peer Review History*

*Press*

Sincerely,

RolI

Roland Roberts, PhD

Senior Editor,

rroberts@plos.org,

PLOS Biology

DATA POLICY:

Regardless of the method selected, please ensure that you provide the individual numerical values that underlie the summary data displayed in the following figure panels as they are essential for readers to assess your analysis and to reproduce it: Figs 1BCDEF, 2E, 3AEFGH, 4BCE, S1BCEF, S2ACD, S3BDE. NOTE: the numerical data provided should include all replicates AND the way in which the plotted mean and errors were derived (it should not present only the mean/average values).

SPECIES INDICATED IN THE ABSTRACT? 

- Please note that per journal policy, the model system/species studied should be clearly stated in the abstract of your manuscript. 

We require the original, uncropped and minimally adjusted images supporting all blot and gel results reported in an article's figures or Supporting Information files. We will require these files before a manuscript can be accepted so please prepare and upload them now. Please carefully read our guidelines for how to prepare and upload this data: https://journals.plos.org/plosbiology/s/figures#loc-blot-and-gel-reporting-requirements

DATA NOT SHOWN?

REVIEWER'S COMMENTS:

Reviewer #1:

The authors did a great job revising the paper. The work is beautiful and advances the field.

---

## [Editor Report · Decision Letter 2]

14 Nov 2022

Dear Markus,

Thank you for the submission of your revised Research Article "Inorganic sulfur fixation via a new homocysteine synthase allows yeast cells to cooperatively compensate for methionine auxotrophy" for publication in PLOS Biology. On behalf of my colleagues and the Academic Editor, Mark Siegal, I'm pleased to say that we can in principle accept your manuscript for publication, provided you address any remaining formatting and reporting issues. These will be detailed in an email you should receive within 2-3 business days from our colleagues in the journal operations team; no action is required from you until then. Please note that we will not be able to formally accept your manuscript and schedule it for publication until you have completed any requested changes.

Sincerely, 

Roli

Senior Editor

PLOS Biology

rroberts@plos.org